# Variational Bayesian Algorithms for Maneuvering Target Tracking with Nonlinear Measurements in Sensor Networks

**DOI:** 10.3390/e25081235

**Published:** 2023-08-18

**Authors:** Yumei Hu, Quan Pan, Bao Deng, Zhen Guo, Menghua Li, Lifeng Chen

**Affiliations:** 1Xi’an Aeronautics Computing Technique Research Institute, AVIC, Xi’an 710069, China; dengbao15@sina.com (B.D.); lmhua1@126.com (M.L.); 2School of Automation, Northwestern Polytechnical University, Xi’an 710072, China; 3Key Laboratory of Information Fusion Technology, Ministry of Education, Xi’an 710072, China; 4System Design Institute of Hubei Aerospace Technology Academy, Wuhan 430040, China; guozhennpu@126.com; 5Department of Precision Instrument, Tsinghua University, Beijing 100084, China; chenlf22@mails.tsinghua.edu.cn

**Keywords:** distributed fusion, nonlinear estimation, variational Bayesian optimization, natural gradient, simultaneous perturbation stochastic approximation, Kullback–Leibler divergence

## Abstract

The variational Bayesian method solves nonlinear estimation problems by iteratively computing the integral of the marginal density. Many researchers have demonstrated the fact its performance depends on the linear approximation in the computation of the variational density in the iteration and the degree of nonlinearity of the underlying scenario. In this paper, two methods for computing the variational density, namely, the natural gradient method and the simultaneous perturbation stochastic method, are used to implement a variational Bayesian Kalman filter for maneuvering target tracking using Doppler measurements. The latter are collected from a set of sensors subject to single-hop network constraints. We propose a distributed fusion variational Bayesian Kalman filter for a networked maneuvering target tracking scenario and both of the evidence lower bound and the posterior Cramér–Rao lower bound of the proposed methods are presented. The simulation results are compared with centralized fusion in terms of posterior Cramér–Rao lower bounds, root-mean-squared errors and the 3σ bound.

## 1. Introduction

There has been much interest in recent years in the use of sensor networks as opposed to single sensors in the fields of target tracking [1], intelligent transportation [2], environmental monitoring [3], spacecraft navigation [4], etc. Multi-sensor nodes can provide greater spatial coverage and, by cooperation, effectively complement the limitations of a single sensor, potentially resulting in improving estimation and fusion performance. Along with the rapid development of sensor network technologies, the two aspects of fusion architecture and state estimation optimization have been studied in the past few years [5,6,7].

Sensor network fusion can be classified into three categories, i.e., centralized, decentralized, and distributed architectures [6,8]. Schematically, examples of different fusion architectures are shown in Figure 1. The topology of sensor networks is, typically, described as an undirected graph where nodes can exchange measurements (or estimation) from neighbors via a bidirectional edge or a directed graph where nodes only can deliver measurements (or estimation) in a fixed direction. Broadcast communications can be single-hop or multi-hop [9]. Here, we only consider the sensor network with undirected topology and single-hop communications. The yellow circles and blue circles describe fusion centers and sensor nodes, respectively.

In the centralized architecture, information fusion takes place after the local sensor measurements are delivered into the fusion center. This architecture can make full use of measurements if the communication bandwidth is high enough to accommodate transmission of all sensor measurements to the fusion center, leading to theoretically optimal fusion [10]. As a result, it is generally utilized as a benchmark for performance comparison and evaluation of the other fusion architectures that will be discussed here. However, there are several inherent problems that need to be taken into account in considering a centralized fusion architecture. One problem is the measurement delay resulting from communication or sensor sampling rate [11,12,13]. Another is that the centralized architecture will, typically, result in a trade-off between timely fusion accuracy performance, and the requirements of communication bandwidth and computational costs because of the broadcast and processing of measurements from all sensors [6,14]. In addition, this architecture is also generally more sensitive to outliers than other fusion architectures [15], such as the following decentralized one.

In decentralized architecture, sensors are partitioned into several clusters, each of them with a fusion center [16,17] where measurements from neighboring nodes are collected to yield a local fused estimation. The decentralized architecture is distinguished from the centralized one by the multiple fusion centers, which can lead to a spread of computational costs across multiple devices and better robustness [6].

In the distributed fusion architecture, each node individually provides a local estimation by using measurements collected from itself and its neighbors. It could be considered as a special form of the decentralized one. Compared with the other two forms, it has the following advantages: (1) Enhanced scalability and feasibility; it allows relatively easy scaling of the network up or down by adding or subtracting nodes depending on practical applications [18]. (2) Increased robustness and fault tolerance to node failures, especially in harsh situations, such as an underwater environment. (3) Reduced bandwidth requirement; it mitigates communication bottlenecks that might arise, for example, in target detection and tracking systems. (4) Reduced computational cost of the three architectures because some processing such as inverse operations generally needed in estimation are performed in their own individual fusion centers. A significant amount of literature on distributed fusion architectures has been published. Ref. [10] is an early paper describing the basic principles of distributed fusion architecture in target tracking systems. A comprehensive review of the characteristics, advantages and estimation solutions for distributed low-cost sensor networks is presented in [6]. For distributed robust filtering, a variational Bayesian (VB) algorithm with a conjugate-exponential model is proposed in [19]. As mentioned in [20], the problem of distributed detection and tracking over a Doppler-shift sensor network is studied. Consensus methods are used in [21,22] for the problem of uncertain noise statistics of distributed sensor networks. However, distributed fusion architecture also has some negative issues, for example, lack of awareness of the sensor network as a whole.

In the context of fusion architectures, state estimation is also essential for target tracking in sensor networks, especially for a nonlinear system. Exact solutions for the posterior probability density function (PDF) for nonlinear systems are mostly unavailable, resulting in a considerable amount of work on approximations of the posterior. A great number of nonlinear filters, such as the extended Kalman filter (EKF) [23], the unscented Kalman filter (UKF) [24], the Gauss–Hermite Kalman filter (GHKF) [25], the central difference Kalman filter (CDKF) [25], the cubature Kalman filter (CKF) [26] and their variants [27,28], have been proposed under a Gaussian noise assumption both on measurements and system. For non-Gaussian noise models, stochastic estimation methods, such as Markov Chain Monte Carlo [29], and sequential Monte Carlo (also named particle filters (PF)) [30] and variants [31,32] have been the focus of much attention over the last two decades. Since these methods need a large number of particles to ensure nonlinear estimation accuracy, Doucet and his colleagues introduced the Rao–Blackwellized particle filter (RBPF) [33] to marginalize out the linear variables to be solved by an optimal filter, and focused the sampled particles on the remaining variables (nonlinear part). For more information about nonlinear estimation problems, we refer the reader to the review articles [34,35,36] where more comprehensive interpretations have been provided.

In terms of optimization techniques, gradient-based optimization is often used in nonlinear filters to improve the performance of nonlinear estimation. As shown in [37], gradient descent is adopted for smartphone orientation estimation, yielding a quaternion-based Kalman filter algorithm to estimate exercise motion. In [38], a gradient descent iterative nonlinear Kalman is proposed for the problem of random missing outputs. For problems where the objective function, such as the variational evidence lower bound (ELBO) and the Kullback–Leibler Divergence (KLD), need to be maximized or minimized, the gradient method generally requires linearization of the objective function. For nonlinear estimation optimization problems, the gradient can be written as a mean of the samples of interest, yielding a stochastic gradient optimization method [39]. In [40], the authors present a modified particle filter where stochastic gradient is used to minimize the criterion function. Compared with the ordinary gradient, natural gradient (NG) has the advantage of theoretical connections from information geometry. According to Amari’s works [41,42], it has a steepest direction in Riemannian space. As shown in [41], NG comes with a theoretical guarantee of asymptotic optimality and can be used to produce a Fisher-efficient iterative estimator on a statistical manifold. In [43], the NG method for the nonlinear estimation problem is proved to be asymptotically optimal in the sense of the Cramér–Rao bound. The earliest interest in NG can be found in [44,45]. Simultaneous perturbation stochastic approximation (SPSA) [46] is an alternative optimization method which can be use for stochastic search. It is a gradient approximation statistic optimization method, which does not require the linearization of the objective function [46]. In this method, a group of samples of objective functions is sampled to obtain a two-side differential function, so that it can be used to approximate the otherwise intractable gradient of the objective function to be estimated. An application of SPSA for detection of the center of a thermal updraft is presented in [47], resulting in an adaptive autonomous soaring algorithm for multiple unmanned aerial vehicles.

In terms of the iterative optimization methods, among the most common approaches are expectation-maximization (EM) [48] and VB [49,50]. EM realizes estimation optimization by establishing a feedback loop, including an expectation step (E-step) and a maximization step (M-step). In the E-step, the conditional expectation of the likelihood function is calculated according to a given prior and measurement. In the M-step, the conditional expectation is maximized to obtain the estimation of the variables. While the state and parameters are estimated and optimized alternately in the iteration cycle, EM still has problems for large scale models and data. VB differs from EM in that the parameters are stochastic [50], resulting in VB being capable of making a joint distribution of state and parameters, and then being especially suitable for dealing with high-dimensional and large-scale problems via mean field theory. Recent advances in the variational iterative framework can be found in [19,51], in which a unified VB approach is provided for the joint estimation of system state and parameters in a target tracking system.

This paper considers the problems of estimation optimization and fusion in networked target tracking systems, and aims to derive a nonlinear estimation, optimization and fusion approach by utilizing VB with the optimization methods of NG and SPSA, achieving a high performance in accuracy. The key contributions of this paper are as follows:Development of a distributed variational fusion framework by utilizing variational mean field theory to approximately partition the joint posterior distribution into several solvable variational distributions under the assumption of independent measurements.Presentation of a novel deterministic nonlinear estimation optimization method to maximize the distributed ELBO using NG, based on linearization to approximate the posterior distributions closely, producing the DVBKF-NG algorithm which yields closed form nonlinear state estimation with the associated covariance for the sensor network.Presentation of a novel stochastic estimation optimization for the VB framework by using SPSA and deriving the stochastic gradient estimation of ELBO, thus producing an iterative filter, i.e., DVBKF-SPSA.Demonstrate performance metrics: distributed ELBO and the posterior Cramér–Rao lower bound (PCRLB) for these algorithms over different iterations.

The rest of the paper is organized as follows. In Section 2, we introduce the general nonlinear estimation problem in target tracking over a sensor network. A distributed variational Bayesian estimation optimization framework is proposed in Section 3, involving partitioning of the joint measurement likelihood. Under the optimization framework, two distributed iterative nonlinear variational Bayesian Kalman filtering algorithms (DVBKF-NG and DVBKF-SPSA) are presented in Section 4, using NG and SPSA, respectively. In Section 5, we consider two kinds of metrics, i.e., ELBO and PCRLB, to evaluate the performance of the proposed methods. In Section 6, we give an example to verify the proposed methods for maneuvering target tracking over a sensor network. Finally, conclusions are drawn and future work is discussed in Section 7.

## 2. Problem Formulation

Consider a maneuvering target travels in the area covered by a fully connected sensor network with *N* nodes; the measurements $Z_{k}={\left[Z_{1,k},\cdots ,Z_{n,k},\cdots ,Z_{N,k}\right]}^{T}$ with noise covariance $R_{k}$ at time index *k* are collected from the set of sensors, where $Z_{n,k}={\left[z_{n,k},z_{n,k}^{1},\cdots ,z_{n,k}^{l},\cdots ,z_{n,k}^{L}\right]}^{T}$ denotes the measurements from sensor *n* and its neighbors, where $z_{n,k}\in R^{d}$ is the measurement of the *n*th sensor and $z_{n,k}^{l}$ is the measurement from the *l*th neighbor of the *n*th sensor, the sensor index n∈{1,2,⋯,N}, l∈{1,2,⋯,L} and satisfies L≤N−1, *d* denotes the dimension of $z_{n,k}$. We assume that only the single-hop communication between two sensors is considered in this paper. The system model and the measurement model are given as follows, respectively,
(1)$$\begin{array}{cc} \; & x_{k}=f_{k|k-1}\left(x_{k-1}\right)+\omega_{k-1} \end{array}$$
(2)$$\begin{array}{cc} \; & z_{n,k}=h_{n,k}\left(x_{k}\right)+\upsilon_{n,k},\;l\in \left\{1,2,\cdots ,L\right\} \end{array}$$
where $x_{k}\in R^{m}$ is the system state which may include position, velocity and other related quantities, and *m* is the dimension of $x_{k}$. Both the system process $\omega_{k}$ and the measurement noise $\upsilon_{n,k}$ of sensor *n* are assumed to be mutually independent and zero-mean Gaussian noises with covariances of $Q_{k-1}$ and $R_{n,k}$, respectively. And $f_{k|k-1}\left(\cdot \right)$ and $h_{n,k}\left(\cdot \right)$ denote the state transfer function and measurement function, respectively.

What we want is to estimate the system state based on the collection of sensor measurements, which is the process of inferring the system state of interest using measurements $Z_{k}$, usually, under the Minimum Mean Square Error (MMSE) criterion, i.e.,
(3)$$\begin{array}{cc} x_{k|k} & ≜E_{p}\left[x_{k}|Z_{k}\right]=\int x_{k}p\left(x_{k}|Z_{k}\right)dx_{k} \end{array}$$
(4)$$\begin{array}{cc} P_{k|k} & ≜E_{p}\left[\left(x_{k}-x_{k|k}\right){\left(x_{k}-x_{k|k}\right)}^{T}|Z_{k}\right]=\int \left(x_{k}-x_{k|k}\right){\left(x_{k}-x_{k|k}\right)}^{T}p\left(x_{k}|Z_{k}\right)dx_{k} \end{array}$$
where $x_{k|k}$ and $P_{k|k}$ are the state estimation and the associated error covariance, and $E_{p}\left[\cdot \right]≜E_{p\left(x_{k}|z_{k}\right)}\left[\cdot \right]$. The posterior distribution $p\left(x_{k}|Z_{k}\right)$ is expressed as
(5)$$p\left(x_{k}|Z_{k}\right)=\frac{p\left(Z_{k}|x_{k}\right)p\left(x_{k}|Z_{k-1}\right)}{\int p\left(x_{k},Z_{k}\right)dx_{k}}$$
where $p\left(x_{k}|Z_{k-1}\right)$ is the a priori distribution and $p\left(Z_{k}|x_{k}\right)$ is the likelihood of measurement $Z_{k}$ with given $x_{k}$. Under Gaussian assumptions, the predicted density $p\left(x_{k}|Z_{k-1}\right)$ is given by
(6)$$p\left(x_{k}|Z_{k-1}\right)∼N\left(x_{k}|f_{k|k-1}\left(x_{k-1|k-1}\right),F_{k|k-1}P_{k-1|k-1}F_{k|k-1}^{T}+Q_{k-1}\right)$$
where $F_{k|k-1}=\frac{\partial f_{k}\left(x\right)}{\partial x}{|}_{x=x_{k-1|k-1}}$. For nonlinear systems, computing the integral $\int p\left(x_{k},z_{k}\right)dx_{k}$ in the denominator of (Equation 5) is in general difficult and the optimal approximation is needed. In this paper, we are interested to use the variational Bayesian method to approximate the a posteriori distribution and thus solve the nonlinear estimation problem of maneuvering target tracking in the described sensor network.

## 3. Distributed Variational Bayesian Estimation Optimization Framework

Given measurement $Z_{k}$ collected from sensors, the principle of VB is given as
(7)$$\begin{array}{c} \log p\left(Z_{k}\right)=L\left(\psi_{k}\right)+D_{\mathrm{KL}}\left[q\left(x_{k}|\psi_{k}\right)\left|\right|p\left(x_{k}|Z_{k}\right)\right] \end{array}$$
and
(8)$$\begin{array}{cc} \; & L\left(\psi_{k}\right)=\int q\left(x_{k}|\psi_{k}\right)\log\frac{p(Z_{k},x_{k})}{q(x_{k}|\psi_{k})}dx_{k} \end{array}$$
(9)$$\begin{array}{cc} \; & D_{\mathrm{KL}}\left[q\left(x_{k}|\psi_{k}\right)\left|\right|p\left(x_{k}|Z_{k}\right)\right]=-\int q\left(x_{k}|\psi_{k}\right)\log\frac{p(x_{k}|Z_{k})}{q(x_{k}|\psi_{k})}dx_{k} \end{array}$$
where $q(x_{k}|\psi_{k})$ is a variational distribution with parameter $\psi_{k}$, $L(\psi_{k})$, and $D_{\mathrm{KL}}\left[q\left(x_{k}|\psi_{k}\right)\left|\right|p\left(x_{k}|Z_{k}\right)\right]$ are the ELBO and the KLD between $q(x_{k}|\psi_{k})$ and $p(x_{k}|Z_{k})$, respectively. What we wish to do is to approximate the $p(x_{k}|Z_{k})$ as closely as possible by minimizing the KLD. Actually, the approximation can be considered as a moving variational distribution along a chosen search direction iteratively to the position of the posterior distribution in a statistical manifold. However, it is difficult to seek out an appropriate method to achieve the approximation, because the variational distribution is usually under-parameterized and not sufficiently flexible to capture the true posterior [52]. As a result, the ELBO is considered to be maximized since maximizing ELBO is equivalent to minimizing KLD from (Equation 7), i.e.,
(10)$$\begin{array}{cc} \log p(Z_{k}) & \geq L\left(\psi_{k}\right)=E_{q}\left[\log p(Z_{k}|x_{k})\right]-D_{\mathrm{KL}}\left[q\left(x_{k}|\psi_{k}\right)\left|\right|p\left(x_{k}\right)\right]. \end{array}$$

We assume that measurements from all sensors are mutually independent; the term $E_{q}\left[\log p(Z_{k}|x_{k})\right]$ in (Equation 10) can be partitioned in distributed fusion architecture as follows
(11)$$E_{q}\left[\log p(Z_{k}|x_{k})\right]=\sum_{n=1}^{N}E_{q}\left[\log p(Z_{n,k}|x_{k})\right]$$
where the measurements $Z_{n,k}$ with associated noise variance $R_{n,k}$ are collected from sensor *n* and its neighbors, where
(12)$$\left\{\begin{array}{c} Z_{n,k}={\left[z_{n,k}^{T},{\left(z_{n,k}^{1}\right)}^{T},\cdots ,{\left(z_{n,k}^{l}\right)}^{T},\cdots ,{\left(z_{n,k}^{L}\right)}^{T}\right]}^{T}\\ p\left(Z_{n,k}|x_{k}\right)=N\left(Z_{n,k};H_{n,k}\left(x_{k}\right),R_{n,k}\right)\\ H_{n,k}\left(\cdot \right)={\left[h_{n,k}^{T}\left(\cdot \right),\cdots ,{\left(h_{n,k}^{L}\left(\cdot \right)\right)}^{T}\right]}^{T}\\ R_{n,k}=\mathrm{Cov}\left(V_{n,k},\;V_{n,k}^{T}\right)=\mathrm{blkdiag}\left(R_{n,k},R_{n,k}^{1},\cdots ,R_{n,k}^{l},\cdots ,R_{n,k}^{L}\right) \end{array}\right.$$
where $z_{n,k}$ denotes the measurement of sensor *n*, and $z_{n,k}^{l}$ denotes the measurement from the *l*th neighbor of sensor *n*, $V_{n,k}={\left[v_{n,k}^{T},{\left(v_{n,k}^{1}\right)}^{T},\cdots ,{\left(v_{n,k}^{l}\right)}^{T},\cdots ,{\left(v_{n,k}^{L}\right)}^{T}\right]}^{T}$ is the measurement noise of sensor *n* and its neighbors, and $R_{n,k}$ and $H_{n,k}\left(x_{k}\right)$ are the associated noise covariance and measurement matrix, respectively, where $R_{n,k}^{l}=\mathrm{Cov}\left(v_{n,k}^{l},\;{\left(v_{n,k}^{l}\right)}^{T}\right)$ denotes the variance of the noise $v_{n,k}^{l}$ from the *l*th neighbor of sensor *n*. The math symbol blkdiag(·) denotes a block diagonal matrix created by aligning input matrices.

For the (i+1)th variational iteration, we can take the *i*th variational distribution $q(x_{k}|\psi_{k}^{i})$ as the prior; the optimized ELBO can be given as
(13)$$\begin{array}{c} L(\psi_{k}^{*})=\sum_{n=1}^{N}E_{q}\left[\log p(Z_{n,k}|x_{k})\right]-D_{\mathrm{KL}}\left[q\left(x_{k}|\psi_{k}\right)\left|\right|q\left(x_{k}|\psi_{k}^{i}\right)\right]. \end{array}$$
We note the definition $\psi_{k}≜\left(x_{k|k},\;P_{k|k}\right)$ in this paper, and wish to update state estimation $x_{k|k}$ and the associated error covariance $P_{k|k}$ in each iteration by maximizing the above ELBO. In the following section, we present two distributed iterative variational Bayesian Kalman filters for maneuvering target tracking by using NG and SPSA, respectively.

## 4. Distributed Iterative Variational Bayesian Kalman Filters over Sensor Network

In this section, with the assumption that the measurements are mutually independent, we present alternative distribution variational Bayesian Kalman filtering algorithms (DVBKF) via NG and SPSA.

### 4.1. NG-Based DVBKF

NG calculated by using information geometry (generally KLD linearization) has a steepest direction in Riemannian space. According to Amari’s works [41,42], we present the NG of objective function $L(\psi_{k})$ with respect to parameter $\psi_{k}$ as follows. Set $\Delta \psi_{k}≜\psi_{k}-\psi_{k}^{i}\to 0$, and  (Equation 13) can be rewritten as
(14)$$\psi_{k}^{*}=\underset{\Delta \psi_{k}\to 0}{\arg\max}\left\{\sum_{n=1}^{N}\nabla_{\psi_{k}}E_{q}\left[\log p\left(Z_{n,k}|x_{k}\right)\right]\Delta \psi_{k}-{\left(\Delta \psi_{k}\right)}^{T}F_{\psi_{k}^{i}}\Delta \psi_{k}\right\}\;$$
where $F_{\psi_{k}^{i}}$ is Fisher information and presented as
(15)$$F_{\psi_{k}^{i}}\approx \nabla_{\psi_{k}^{i}}^{2}D_{\mathrm{KL}}\left[q\left(x_{k}|\psi_{k}\right)∥q\left(x_{k}|\psi_{k}^{i}\right)\right].$$
The proof can be seen in [53]. After computing the partial derivative of the right side of (Equation 14) and setting it equal to 0, the NG of the ELBO at the *i*th iteration in distributed architecture is expressed as
(16)$${\tilde{\nabla }}_{\psi_{k}^{i}}=F_{\psi_{k}^{i}}^{-1}\sum_{n=1}^{N}\nabla_{\psi_{k}^{i}}E_{q}\left[\log p\left(Z_{n,k}|x_{k}\right)\right].$$

The NG ${\tilde{\nabla }}_{\psi_{k}^{i}}$ is the direction in which the increase of ELBO is greatest [54]; that means it has the greatest descent or ascent at each iteration in statistical manifold space, and can move the variational distribution to approximate the posterior fastest. At this point, the optimal $\psi_{k}$ is presented as  
(17)$$\psi_{k}^{i+1}=\psi_{k}^{i}+{\tilde{\nabla }}_{\psi_{k}^{i}}$$

It is observed that (Equation 17) is a general expression of the iterative update of parameter $\psi_{k}$. In terms of the update of $x_{k|k}^{i+1}$ and $P_{k|k}^{i+1}$, the special forms of Fisher information matrices ($F_{x_{k|k}^{i}}^{-1}$ and $F_{P_{k|k}^{i}}^{-1}$) and the gradients of log-likelihood expectation ($\nabla_{x_{k|k}^{i}}E_{q}\left[\log p\left(Z_{n,k}|x_{k}\right)\right]$ and $\nabla_{P_{k|k}^{i}}E_{q}\left[\log p\left(Z_{n,k}|x_{k}\right)\right]$) with respect to $x_{k|k}$ and $P_{k|k}$ need to be analyzed.

With the Gaussian system assumption, the KLD between two Gaussian distributions $q_{1}∼N\left(\xi_{1};\mu_{1},C_{1}\right)$ and $q_{2}∼N\left(\xi_{2};\mu_{2},C_{2}\right)$ with the same dimension *d* is given as
(18)$$\begin{array}{c} D_{\mathrm{KL}}\left[q_{1}∥q_{2}\right]=\frac{1}{2}\left\{\ln\left(\left|C_{2}\right|{\left|C_{1}\right|}^{-1}\right)+\mathrm{tr}\left(C_{2}^{-1}C_{1}\right)\right.\left.+{\left(\mu_{2}-\mu_{1}\right)}^{T}C_{2}^{-1}\left(\mu_{2}-\mu_{1}\right)-d\right\} \end{array}$$
Combining with (Equation 15), the Fisher information matrices with respect to $x_{k|k}^{i}$ and $P_{k|k}^{i}$ are presented as
(19)$$\begin{array}{c} F_{x_{k|k}^{i}}={\left(P_{k|k}^{i}\right)}^{-1} \end{array}$$
(20)$$\begin{array}{c} F_{P_{k|k}^{i}}\approx \frac{1}{2}{\left(P_{k|k}^{i}\right)}^{-1}⊗{\left(P_{k|k}^{i}\right)}^{-1}. \end{array}$$
The gradient of log-likelihood expectation with respect to $x_{k|k}^{i}$ and $P_{k|k}^{i}$ of sensor *n* are presented as
(21)$$\begin{array}{cc} \nabla_{x_{k|k}^{i}}E_{q}\left[\log p\left(Z_{n,k}|x_{k}\right)\right]\approx & H_{n,x_{k|k}^{i}}^{T}R_{n,k}^{-1}\left(Z_{n,k}-H_{n,k}\left(x_{k|k}^{i}\right)\right) \end{array}$$
(22)$$\begin{array}{cc} \nabla_{P_{k|k}^{i}}E_{q}\left[\log p\left(Z_{n,k}|x_{k}\right)\right]\approx & -\frac{1}{2}H_{n,x_{k|k}^{i}}^{T}R_{n,k}^{-1}H_{n,x_{k|k}^{i}} \end{array}$$
where $H_{n,x_{k|k}^{i}}=\frac{\partial H_{n,k}\left(x_{k}\right)}{\partial x_{k|k}}{|}_{x_{k|k}=x_{k|k}^{i}}$ denotes the Jacobian matrix of the measurement matrix of the *n*th sensor.

Recalling the iterative optimization forms in (Equation 16) and (Equation 17), the distributed iterative state estimation $x_{k|k}^{i+1}$ and the associated covariance $P_{k|k}^{i+1}$ in DVBKF-NG are, respectively, given by
(23)$$\begin{array}{c} x_{k|k}^{i+1}=x_{k|k}^{i}+P_{k|k}^{i}\sum_{n=1}^{N}H_{n,x_{k|k}^{i}}^{T}R_{n,k}^{-1}\left(Z_{n,k}-H_{n,k}\left(x_{k|k}^{i}\right)\right) \end{array}$$
(24)$$\begin{array}{c} P_{k|k}^{i+1}=P_{k|k}^{i}\left(I-\sum_{n=1}^{N}H_{n,x_{k|k}^{i}}^{T}R_{n,k}^{-1}H_{n,x_{k|k}^{i}}P_{k|k}^{i}\right). \end{array}$$

The iterative optimization process of DVBKF-NG is summarized in Algorithm 1. We make the following remarks:The update of $x_{k|k}^{i}$ is preconditioned by $P_{k|k}^{i}$, which produces an adaptive movement to the posterior PDF along the direction of NG.Clearly, Equation (24) shows that $P_{n,k|k}^{i+1}\leq P_{n,k|k}^{i}$, which means that estimation error covariance decreases gradually at each iteration. Additionally, since $H_{n,x_{k|k}^{i}}^{T}R_{n,k}^{-1}H_{n,x_{k|k}^{i}}$ is the expectation of the Hessian, $P_{n,k|k}^{i}$ has a quadratic convergence.To make the algorithm adaptive, the relative estimation error $e_{r}$ can be used for judging the iteration termination.
(25)$$e_{r}=|\frac{x_{k|k}^{i+1}-x_{k|k}^{i}}{x_{k|k}^{i}}|\leq \epsilon$$
where ϵ is a small positive number which can be chosen according to practical scenarios.
**Algorithm 1** The DVBKF-NG algorithm1:Initialize state estimation $x_{1|1}$, estimation error covariance $P_{1|1}$, the number of iterations Iter;2:Compute one-step predicted state$x_{k|k-1}=f_{k|k-1}\left(x_{k-1|k-1}\right)$;3:Compute predicted state error covariance$P_{k|k-1}=F_{k|k-1}P_{k-1|k-1}F_{k|k-1}^{T}+Q_{k}$;4:Let i=1, and $x_{k|k}^{1}=x_{k|k-1}$, $P_{k|k}^{1}=P_{k|k-1}$.5:**for** each iteration i=1:Iter and $e_{r}\leq \epsilon$ **do**6:   Compute Fisher information matrices with respect to $x_{k|k}^{i+1}$ and $P_{k|k}^{i+1}$ by (Equation 19) and (Equation 20).7:   Compute the gradients of log-likelihood expectation of each sensor with respect to $x_{k|k}^{i+1}$ and $P_{k|k}^{i+1}$ by  (Equation 21) and (Equation 22), respectively.8:   Compute the iterative state estimation $x_{k|k}^{i+1}$ and the associated error covariance $P_{k|k}^{i+1}$ by (Equation 23) and (Equation 24), respectively.9:   Compute relative error $e_{r}$ by (Equation 25).10:**end for**11:Output $x_{k|k}=x_{k|k}^{i+1}$, $P_{k|k}=P_{k|k}^{i+1}$.

### 4.2. SPSA-Based DVBKF

SPSA is a statistical optimization method for gradient approximation, which does not require the full knowledge of the objective function being minimized (or maximized) and parameters being optimized [46]. In this method, a group of samples of objective function $L(\psi_{k})$ is sampled as $Y\left(\psi_{k}\right)=L\left(\psi_{k}\right)+\zeta$ to obtain a two-side differential function, where ζ is a random infinitesimal perturbation. Then, the two-side infinitesimal function is computed by
(26)$$dY\left(\psi_{k}^{i}\right)=Y\left(\psi_{k}+c_{\psi_{k}^{i}}\Delta_{\psi_{k}^{i}}\right)-Y\left(\psi_{k}-c_{\psi_{k}^{i}}\Delta_{\psi_{k}^{i}}\right).$$
where $\Delta_{\psi_{k}^{i}}$ is random perturbation vector with a Gaussian distribution form and $c_{\psi_{k}^{i}}$ is a small positive number that decreases with *i*. As a result, the estimation of the gradient of objective function $L(\psi_{k})$ can be obtained by the two-side differential. The *m*th component of the gradient estimator at the *i*th iteration is given as follows,
(27)$${\left(\hat{g}\left(\psi_{k}^{i}\right)\right)}_{m}=\frac{Y\left(\psi_{k}+c_{\psi_{k}^{i}}\Delta_{\psi_{k}^{i}}\right)-Y\left(\psi_{k}-c_{\psi_{k}^{i}}\Delta_{\psi_{k}^{i}}\right)}{2c_{\psi_{k}^{i}}{\left(\Delta_{\psi_{k}^{i}}\right)}_{m}}$$
where m∈{1,2,,⋯,M}, *M* is the dimension of $\psi_{k}$. The gradient estimation of $L\left(\psi_{k}\right)$ at the *i*th iteration is presented as
(28)$$\hat{G}\left(\psi_{k}^{i}\right)={\left[{\left(\hat{g}\left(\psi_{k}^{i}\right)\right)}_{1}\;{\left(\hat{g}\left(\psi_{k}^{i}\right)\right)}_{2}\;\cdots \;{\left(\hat{g}\left(\psi_{k}^{i}\right)\right)}_{M}\right]}^{T}=\frac{dY\left(\psi_{k}^{i}\right)}{2c_{\psi_{k}^{i}}}\Lambda_{\psi_{k}^{i}}$$
where $\Lambda_{\psi_{k}^{i}}={\left[{\left(\Delta_{\psi_{k}^{i}}\right)}_{1}^{-1}\;{\left(\Delta_{\psi_{k}^{i}}\right)}_{2}^{-1}\;\cdots \;{\left(\Delta_{\psi_{k}^{i}}\right)}_{M}^{-1}\right]}^{T}$, $\Lambda_{\psi_{k}^{i}}$ is random perturbation vector with multivariate Gaussian distribution form and ${\left(\Delta_{\psi_{k}^{i}}\right)}_{m}$ denotes the *m*th element of $\Lambda_{\psi_{k}^{i}}$. At this point, the $\psi_{k}^{i+1}$ can be updated by
(29)$$\psi_{k}^{i+1}=\psi_{k}^{i}+a_{\psi_{k}^{i}}\hat{G}\left(\psi_{k}^{i}\right)$$
where $a_{\psi_{k}^{i}}$ is a weighted factor.

Taking the distributed variational ELBO $L\left(\psi_{k}\right)$ in (Equation 13) as the objective function, the two-side infinitesimal perturbations of $L\left(\psi_{k}\right)$ with respect to $x_{k|k}^{i}$ and $P_{k|k}^{i}$ are given, respectively, as
(30)$$\begin{array}{cc} L\left(x_{k|k}\pm c_{x_{k|k}^{i}}\Delta_{x_{k|k}^{i}}\right)= & \sum_{n=1}^{N}E_{q}\left[\log p\left(Z_{n,k}|x_{k|k}^{i}\pm c_{x_{k|k}^{i}}\Delta_{x_{k|k}^{i}}\right)\right]\\ \; & -D_{\mathrm{KL}}\left[N\left(x_{k}|x_{k|k}^{i}\pm c_{x_{k|k}^{i}}\Delta_{x_{k|k}^{i}},P_{k|k}\right)\left|\right|N\left(x_{k}|x_{k|k}^{i},P_{k|k}^{i}\right)\right] \end{array}$$
(31)$$\begin{array}{cc} L\left(P_{k|k}\pm c_{P_{k|k}^{i}}\Delta_{P_{k|k}^{i}}\right)= & -\frac{1\pm c_{P_{k|k}^{i}}}{2}\sum_{n=1}^{N}H_{n,x_{k|k}^{i}}^{T}R_{k}^{-1}H_{n,x_{k|k}^{i}}\Delta_{P_{k|k}^{i}}\\ \; & -D_{\mathrm{KL}}\left[N\left(x_{k}|x_{k|k}^{i},P_{k|k}\pm c_{P_{k|k}^{i}}\Delta_{P_{k|k}^{i}}\right)\left|\right|N\left(x_{k}|x_{k|k}^{i},P_{k|k}^{i}\right)\right] \end{array}$$
where the parameters $c_{x_{k|k}^{i}}$ and $c_{P_{k|k}^{i}}$ are the special forms of $c_{\psi_{k}^{i}}$ with respect to $x_{k|k}^{i}$ and $P_{k|k}^{i}$, respectively.

Now, we formulate the random perturbation factors of $\Delta_{x_{k|k}^{i}}$ and $\Delta_{P_{k|k}^{i}}$ by sampling from the Gaussian distribution $x_{k}^{s}∼N\left(x_{k|k}^{i},\;P_{k|k}^{i}\right)$; the mean and covariance of the samples are ${\bar{x}}_{k}^{s}=\frac{1}{S}\sum_{s=1}^{S}x_{k}^{s}$ and $\delta^{2}=\frac{1}{S-1}\sum_{s=1}^{S}\left(x_{k}^{s}-{\bar{x}}_{k}^{s}\right){\left(x_{k}^{s}-{\bar{x}}_{k}^{s}\right)}^{T}$, where $s\in \left\{1,\;2,\;\cdots ,\;S\right\}$. Randomly choose a sample $x_{k}^{s}$; the random perturbation of state $\Delta_{x_{k|k}^{i}}$ is given by
(32)$$\Delta_{x_{k|k}^{i}}=x_{k|k}^{i}-x_{k}^{s}.$$
The associated random perturbation of covariance is given as
(33)$$\Delta_{P_{k|k}^{i}}=\delta^{2}-P_{k|k}^{i}.$$
Recall (Equation 26), the two-side differential $dY\left(\psi_{k}^{i}\right)$ with respect to $x_{k|k}^{i}$ and $P_{k|k}^{i}$ is written as
(34)$$\begin{array}{cc} dY\left(x_{k|k}^{i}\right) & =L\left(x_{k|k}^{i}+c_{x_{k|k}^{i}}\Delta_{x_{k|k}^{i}}\right)-L\left(x_{k|k}^{i}-c_{x_{k|k}^{i}}\Delta_{x_{k|k}^{i}}\right)+\zeta_{x} \end{array}$$
(35)$$\begin{array}{cc} dY\left(P_{k|k}^{i}\right) & =L\left(P_{k|k}^{i}+c_{P_{k|k}^{i}}\Delta_{P_{k|k}^{i}}\right)-L\left(P_{k|k}^{i}-c_{P_{k|k}^{i}}\Delta_{P_{k|k}^{i}}\right)+\zeta_{P}. \end{array}$$
where $\zeta_{x}$ and $\zeta_{P}$ are random infinitesimal perturbations.

It follows that the estimation of the gradient of $L\left(\psi_{k}\right)$ with respect to $x_{k|k}^{i}$ and $P_{k|k}^{i}$ at the *i*th iteration is given by
(36)$$\begin{array}{cc} \hat{G}\left(x_{k|k}^{i}\right)= & \frac{dY\left(x_{k|k}^{i}\right)}{2c_{x_{k|k}^{i}}\Delta_{x_{k|k}^{i}}}\Lambda_{x_{k|k}^{i}} \end{array}$$
(37)$$\begin{array}{cc} \hat{G}\left(P_{k|k}^{i}\right)= & \frac{dY\left(P_{k|k}^{i}\right)}{2c_{P_{k|k}^{i}}\Delta_{P_{k|k}^{i}}}\Lambda_{P_{k|k}^{i}} \end{array}$$
where
$$\begin{array}{cc} \Lambda_{x_{k|k}^{i}}= & {\left[{\left(\Delta_{x_{k|k}^{i}}\right)}_{1}^{-1}\;{\left(\Delta_{x_{k|k}^{i}}\right)}_{2}^{-1}\;\cdots \;{\left(\Delta_{x_{k|k}^{i}}\right)}_{M}^{-1}\right]}^{T}\\ \Lambda_{P_{k|k}^{i}}= & \mathrm{diag}\left[{\left(\Delta_{P_{k|k}^{i}}\right)}_{1,1}^{-1}\;{\left(\Delta_{P_{k|k}^{i}}\right)}_{2,2}^{-1}\;\cdots \;{\left(\Delta_{P_{k|k}^{i}}\right)}_{M,M}^{-1}\right] \end{array}$$
and ${\left(\Delta_{P_{k|k}^{i}}\right)}_{m,m}$ denotes the element in the *m*th row and *m*th column of $\Delta_{P_{k|k}^{i}}$. Therefore, state estimation and the associated covariance are updated by
(38)$$\begin{array}{cc} x_{k|k}^{i+1}= & x_{k|k}^{i}+a_{x_{k|k}^{i}}\hat{G}\left(x_{k|k}^{i}\right) \end{array}$$
(39)$$\begin{array}{cc} P_{k|k}^{i+1}= & P_{k|k}^{i}+a_{P_{k|k}^{i}}\hat{G}\left(P_{k|k}^{i}\right) \end{array}$$
where the parameters of $a_{x_{k|k}^{i}}$ and $a_{P_{k|k}^{i}}$ are weighted factors. Set the same iteration termination as (Equation 25); the DVBKF-SPSA algorithm is summarized in Algorithm 2.
**Algorithm 2** The iterative optimization process in the DVBKF-SPSA algorithm1:Initialize state estimation $x_{1|1}$, estimation error covariance $P_{1|1}$, the number of iterations Iter;2:Compute one-step predicted state$x_{k|k-1}=f_{k|k-1}\left(x_{k-1|k-1}\right)$;3:Compute predicted state error covariance$P_{k|k-1}=F_{k|k-1}P_{k-1|k-1}F_{k|k-1}^{T}+Q_{k}$;4:Let i=1, and $x_{k|k}^{1}=x_{k|k-1}$, $P_{k|k}^{1}=P_{k|k-1}$.5:**for** each iteration i=1:Iter and e≤ϵ **do**6:   Sampling from the Gaussian distribution   $x_{k}^{s}∼N\left(x_{k|k}^{i},\;P_{k|k}^{i}\right)$.7:   Compute stochastic perturbations $\Delta_{x_{k|k}^{i}}$ and $\Delta_{P_{k|k}^{i}}$ by (Equation 32) and (Equation 33), respectively.8:   Compute the two-side infinitesimal perturbations of $L\left(\psi_{k}\right)$ with respect to $x_{k|k}^{i}$ and $P_{k|k}^{i}$ by  (Equation 30) and (Equation 31), respectively.9:   Compute the estimation of gradient $\hat{G}\left(x_{k|k}^{i}\right)$ and $\hat{G}\left(P_{k|k}^{i}\right)$ by (Equation 36) and (Equation 37), respectively.10   Update distributed iterative state estimation $x_{k|k}^{i+1}$ and the associated error covariance $P_{k|k}^{i+1}$ by (Equation 38) and (Equation 39), respectively.11:   Compute relative error $e_{r}$ by (Equation 25).12:**end for**13:Output $x_{k|k}=x_{k|k}^{i+1}$, $P_{k|k}=P_{k|k}^{i+1}$.

## 5. Performance Evaluation

It is often of interest to know how closely the posterior distribution is approximated and how accurately a variable can be estimated. In this section, we present two metrics for the proposed algorithm performance evaluation: One is variational ELBO which is monotonically increasing in iteration index to measure the convergence of variational iteration. The other is PCRLB which provides a lower bound on the mean square error of system state estimation [27]. In this section, we present the general forms of ELBO and PCRLB both of DVBKF-NG and DVBKF-SPSA.

### 5.1. Performance in ELBO

From (Equation 10) and (Equation 11), the iterative ELBO of distributed architecture can be rewritten as
(40)$$\begin{array}{c} L\left(\psi_{k}^{i}\right)=\sum_{n=1}^{N}E_{q}\left[\log p(Z_{n,k}|x_{k})\right]+E_{q}\left[\log p\left(x_{k}\right)\right]-E_{q}\left[\log q\left(x_{k}|\psi_{k}^{i}\right)\right] \end{array}$$
After computing the log-likelihood expectations in (Equation 40) under the Gaussian assumption, we present the iterative ELBO of DVBKF (see the derivation in Appendix A) over the sensor network, as follows
(41)$$\begin{array}{cc} L\left(\psi_{k}^{i}\right)=-\frac{1}{2}\{ & \sum_{n=1}^{N}\mathrm{tr}\left(R_{n,k}^{-1}\left(\left(Z_{n,k}-H_{n,x_{k|k}^{i}}x_{k|k}^{i}\right){\left(Z_{n,k}-H_{n,x_{k|k}^{i}}x_{k|k}^{i}\right)}^{T}+H_{n,x_{k|k}^{i}}P_{k|k}^{i}H_{n,x_{k|k}^{i}}^{T}\right)\right)\\ \; & +\mathrm{tr}\left(P_{k|k-1}^{-1}\left(\left(x_{k|k}^{i}-x_{k|k-1}\right){\left(x_{k|k}^{i}-x_{k|k-1}\right)}^{T}+P_{k|k}^{i}\right)\right)\\ \; & \left.+\log\left(|P_{k|k-1}\left|\right|P_{k|k}^{i}{|}^{-1}\prod_{n=1}^{N}\left|R_{n,k}\right|\right)+\sum_{n=1}^{N}D_{n,z}\log\left(2\pi \right)-D_{x}\right\} \end{array}$$
in which $D_{n,z}$ and $D_{x}$ are the dimensions of $Z_{n,k}$ and $x_{k}$, and $H_{n,x_{k|k}^{i}}=\frac{\partial H_{n,k}\left(x_{k}\right)}{\partial x_{k}}{|}_{x_{k}=x_{k|k}^{i}}$.

### 5.2. Performance in PCRLB

The PCRLB provides a theoretical lower bound for the estimation problem under a distributed Bayesian framework. It has the following defined form [55]
(42)$$P_{k+1|k+1}≜E_{q(x_{k}|\psi_{k})}\left[\left(x_{k}-x_{k|k}\right){\left(x_{k}-x_{k|k}\right)}^{T}\right]\geq J_{k+1}^{-1}$$
where $J_{k+1}$ is the posterior Fisher information matrix, recursively computed by
(43)$$J_{k+1}=D_{k}^{22}-D_{k}^{21}{\left(J_{k}+D_{k}^{11}\right)}^{-1}D_{k}^{12}.$$
The terms in (Equation 43) can be expressed by
(44)$$\begin{array}{cc} D_{k}^{11}= & E\left[-\nabla_{x_{k}}\nabla_{x_{k}}^{T}\log p\left(x_{k+1}|x_{k}\right)\right]\\ D_{k}^{12}= & E\left[-\nabla_{x_{k}}\nabla_{x_{k+1}}^{T}\log p\left(x_{k+1}|x_{k}\right)\right]\\ D_{k}^{21}= & E\left[-\nabla_{x_{k+1}}\nabla_{x_{k}}^{T}\log p\left(x_{k+1}|x_{k}\right)\right]={\left[D_{k}^{12}\right]}^{T}\\ D_{k}^{22}= & E\left[-\nabla_{x_{k+1}}\nabla_{x_{k+1}}^{T}\log p\left(x_{k+1}|x_{k}\right)\right]+E\left[-\nabla_{x_{k+1}}\nabla_{x_{k+1}}^{T}\log p\left(Z_{k+1}|x_{k+1}\right)\right]. \end{array}$$
From (Equation 44), we can know that the terms of $D_{k}^{11}$, $D_{k}^{12}$ and $D_{k}^{21}$ are related only to the system model and irrelated to the fusion architecture of the sensor network. With the assumption of independent measurements, $\log p\left(Z_{k+1}|x_{k+1}\right)=\sum_{n=1}^{N}\log p\left(Z_{n,k+1}|x_{k+1}\right)$.
(45)$$\begin{array}{cc} D_{k}^{22}= & E\left[-\nabla_{x_{k+1}}\nabla_{x_{k+1}}^{T}\log p\left(x_{k+1}|x_{k}\right)\right]+\sum_{n=1}^{N}E\left[-\nabla_{x_{k+1}}\nabla_{x_{k+1}}^{T}\log p\left(Z_{n,k+1}|x_{k+1}\right)\right]. \end{array}$$

After computing the gradients in (Equation 44) and (Equation 45) by linearizing $h_{k}\left(x_{k}\right)$ and $f_{k|k-1}\left(x_{k-1|k-1}\right)$, the PCRLB (see the derivation in Appendix B) of DVBKF has the following form
(46)$$\begin{array}{cc} P_{k|k}^{-1}= & {\left(Q_{k-1}+F_{k|k-1}P_{k-1|k-1}F_{k|k-1}^{T}\right)}^{-1}+\sum_{n=1}^{N}{\left(H_{n,x_{k|k}^{i}}\right)}^{T}R_{n,k}^{-1}H_{n,x_{k|k}^{i}}. \end{array}$$

### 5.3. Remarks

From (Equation 40) and (Equation 41), it is observed that the ELBO values both of DVBKF-NG and DVBKF-SPSA are related to measurements, prior, variational distribution and the number of iterations. Generally, it is assumed that the prior of a given objective system is known. Therefore, choosing an appropriate form of parameterized variational distribution, increasing the number of iterations and providing more measurement are of the essence to maximize the ELBO and lead a close approximation of the posterior. However, the balance between computation cost and accuracy should be considered according to practical applications.The PCRLB is an important metric to evaluate the accuracy of estimation algorithms. On the one hand, it is determined by models of dynamic systems and measurement systems. On the other hand, similarly to the ELBO, the PCRLB values both of DVBKF-NG and DVBKF-SPSA are still related to the number of iterations and the amount of measurement.The two metrics are inextricably linked with each other: The former is the means and the latter is the goal. ELBO maximization means approximating the posterior distribution closely by an iterative variational distribution, which can lead the PCRLB to a lower trend.

## 6. Numerical Simulation

In this section, we present a scenario of 2-D maneuvering target tracking over a sensor network with Doppler-only measurement to illustrate the performance of DVBKF-NG and DVBKF-SPSA. We also present the NG-based and SPSA-based optimizations for centralized fusion architecture, named CVBKF-NG and CVBKF-SPSA, which can be utilized, respectively, as benchmarks corresponding to DVBKF-NG and DVBKF-SPSA for performance comparison.

From the system observability theory, we can know that target state with Doppler-only measurement only can be observable after collecting measurements from at least three Doppler sensors with different fixed locations. Thus, the measurements from neighbors along with the measurements of local sensors are used to estimate variables in distribution fusion architecture.

### 6.1. Performance Metrics

The estimation performance of the proposed algorithms is measured by root-mean-squared error (RMSE), 3σ rule and the mean running overhead with 1000 Monte Carlo simulations. The RMSE in range $R_{k}^{\mathrm{RMSE}}$, RMSE in radical velocity $V_{k}^{\mathrm{RMSE}}$, 3σ in range $R_{k}^{3\sigma }$, 3σ in radical velocity $V_{k}^{3\sigma }$ and mean running time $T_{\mathrm{mean}}$ are given, respectively, as follows
$$\begin{array}{cc} \; & R_{k}^{\mathrm{RMSE}}=\sqrt{\frac{1}{M}\sum_{m=1}^{M}\left|\right|x_{k}^{p}-x_{k|k}^{p}{\left|\right|}^{2}}\\ \; & V_{k}^{\mathrm{RMSE}}=\sqrt{\frac{1}{M}\sum_{m=1}^{M}\left|\right|x_{k}^{v}-x_{k|k}^{v}{\left|\right|}^{2}}\\ \; & R_{k}^{3\sigma }=3\sqrt{\frac{1}{M}\sum_{m=1}^{M}{\left(\left|\right|x_{k}^{p}\left|\right|-\left|\right|x_{k|k}^{p}\left|\right|\right)}^{2}}\\ \; & V_{k}^{3\sigma }=3\sqrt{\frac{1}{M}\sum_{m=1}^{M}{\left(\left|\right|x_{k}^{v}\left|\right|-\left|\right|x_{k|k}^{v}\left|\right|\right)}^{2}}\\ \; & T_{\mathrm{mean}}=\frac{1}{MK}\sum_{m=1}^{M}\sum_{k=1}^{K}t_{m,k} \end{array}$$
where $x_{k}={\left[x_{k}^{p}\;x_{k}^{v}\right]}^{T}$, $x_{k}^{p}={\left[x\;y\right]}^{T}$ and $x_{k}^{v}={\left[\dot{x}\;\dot{y}\right]}^{T}$ denote the true position vector and velocity vector of the target, where $x_{k|k}^{p}$ and $x_{k|k}^{v}$ denote the associated estimation, respectively, ∥·∥ denotes the Euclidean norm and $t_{m,k}$ is the running time at the *k*th estimation in the *m*th Monte Carlo simulation. The values of the parameters in the proposed algorithms are given in Table 1.

### 6.2. Simulation Setup

In this scenario, we consider a sensor network which consists of 20 Doppler-only sensors located in a square of 120m×120m randomly, as shown in Figure 2. The communication capability of each sensor node is 50m. The target is maneuvering with dynamic multiple models $F_{k}^{j}$, j∈{1,2} given as follows.
(47)$$F_{k}^{j}=\left[\begin{array}{cccc} 1 & 0 & \sin\left(\theta_{j}T\right)/\theta_{j} & (\cos\left(\theta_{j}T\right)-1)/\theta_{j}\\ 0 & 1 & \left(1-\cos\left(\theta_{j}T\right)\right)/\theta_{j} & \sin\left(\theta_{j}T\right)/\theta_{j}\\ 0 & 0 & \cos\left(\theta_{j}T\right) & -\sin\left(\theta_{j}T\right)\\ 0 & 0 & \sin\left(\theta_{j}T\right) & \cos\left(\theta_{j}T\right) \end{array}\right]$$
(48)$$\left\{\begin{array}{cc} j=1, & k\in \{[1,14),\;[18,30),\;[34,49),\;[53,71),\;[75,90)\};\\ j=2, & k\in \{[14,18),\;[30,34),\;[49,53),\;[71,75)\}. \end{array}\right.$$
where the turn rates are $\theta_{1}=-9.8N_{1}/v_{k}$ and $\theta_{2}=-\theta_{1}$, $N_{1}=0.2$ is the overloads of target maneuvering and the scan period is T=0.2 s. In this manuscript, the target state is represented as $x_{k}={\left[x_{k}^{p}\;x_{k}^{v}\right]}^{T}$, where $x_{k}^{p}={\left[x\;y\right]}^{T}$ and $x_{k}^{v}={\left[\dot{x}\;\dot{y}\right]}^{T}$ denote the true position and velocity of the target. Both measurements $z_{k}$ and $x_{k}$ are defined in a macro sense, not as one state or one measurement. In the simulation, the state of velocity is updated in the coordinates. The initial state estimation is $x_{1|1}={\left[-40\;m\;\;-40\;m\;\;3.5\;m/s\;\;0\;m/s\right]}^{T}$ and its estimated error covariance is given as $P_{1|1}=\mathrm{diag}\left([\delta_{1}^{2}\;\;\delta_{1}^{2}\;\;\delta_{2}^{2}\;\;\delta_{2}^{2}]\right)$, where $\delta_{1}=\sqrt{1.5}\;m$ and $\delta_{2}=\sqrt{0.03}\;m/s$. System noise covariance is given as $Q_{k}=\mathrm{diag}\left(\left[q_{1}^{2}\;\;q_{1}^{2}\;\;q_{2}^{2}\;\;q_{2}^{2}\right]\right)$, where $q_{1}=0.01\;m$ and $q_{2}=0.01\;m/s$.

Sensor measurements are described by a nonlinear equation of Doppler shift between the target and each of the sensors, and measurement noise covariances are time-varying because of target motion. The measurement function and measurement noise covariances are given by [56,57]
(49)$$h_{k}^{\left(j\right)}\left(x_{k}\right)=\frac{2\left(x_{k}^{p}-S^{n}\right)\left|\right|x_{k}^{v}\left|\right|}{∥x_{k}^{p}-S^{n}∥}$$
(50)$$R_{n,k}^{j}=\frac{3}{\pi^{2}T_{d}^{2}R_{\mathrm{SNR}}}$$
Equations (49) and (50) represent the measurement model of the sensors, where (49) is the mapping from state to measurement and represents the Doppler shift between the target and each of the sensors. Since the radial velocity of the target is related to Doppler shift, we update it by using the velocity in the coordinates. Equation (Equation 50) represents the associated time-varying measurement variance because of target motion, where $T_{d}=1$ μs is the pulse Doppler waveform width, $R_{\mathrm{SNR}}$ is the signal-to-noise ratio (SNR) and $R_{\mathrm{SNR}}=\frac{P_{e}}{P_{n}}$. For a given transmitted waveform with unit energy, the energy of the received signal $P_{e}=\frac{P_{t}GA_{e}\delta }{{\left(4\pi \right)}^{2}r^{4}}$, where *r* is radar radius, *G* and δ are the radar antenna and the cross-section area, respectively, and $P_{t}$ and $A_{e}$ are the echo power and the effective receiving area of radar antenna, respectively. The relationship between measurement noise covariance and the range from radar to target is presented in Appendix C for a Gaussian noise $P_{n}=\frac{kT_{s}}{2}$, where $T_{s}=290\;K$ and $k=1.3806\times 10^{-23}\;J/K$ are the temperature in degrees Kelvin and the Boltzmann constant, respectively.

Figure 3a shows the number of neighbors of each sensor node. It is clearly the 4th sensor has the most neighbors, numbering 12, then the 1st, 6th, 12th and 14th sensors, and the 17th sensor has the least number of neighbors. Besides the proposed iterative algorithms, we use the interacting multiple model (IMM) method to achieve the model transfer in simulation. In distributed architecture, the sensor with best estimation accuracy is chosen for output. From Figure 3b, we can observe that sensor 4 has the largest number of outputs, then sensors 6, 14, 12 and 1. To a certain degree, this reflects that multi-sensor measurements have the advantage of improving accuracy.

### 6.3. Simulation Results and Analysis

Figure 4a,b shows the PCRLB and RMSE curves in range and radical velocity, respectively. From the view of fusion architecture, centralized fusion architecture has lower RMSE curves and PCRLB curves than the distributed one. Namely, CEKF, CVBKF-NG and CVBKF-SPSA are, respectively, better than the associated distributed DEKF, DVBKF-NG and DVBKF-SPSA, since the state estimation is updated by using the measurements from the sensor and its neighbor in distributed architecture. In contrast, the measurements collected from all sensors are used to update state estimation in centralized fusion architecture.

From the view of optimization methods, we can observe that the RMSE curves of DVBKF-NG and DVBKF-SPSA are better than DEKF which does not use any optimization methods. Besides iterative linearization, random perturbation sampling which can capture more information from nonlinear measurement is used in DVBKF-SPSA. As a result, the measurement is utilized effectively to improve the accuracy performance. As shown in Figure 4, the RMSE curves obtained by using SPSA are lower than those obtained by NG optimization both in range and radical velocity. However, from Figure 4b, it is also found that the proposed algorithms and comparison algorithms in distributed architecture are more sensitive than those in the centralized one when the dynamic model transfers. The comparisons of the PCRLB and RMSE of estimated position and velocity in coordinates are given in Figure 5 and Figure 6, respectively.

More quantitative RMSE comparison in centralized architecture and distributed architecture can be seen in Table 2 and Table 3, from which it can be seen that the RMSE curves of the proposed algorithms DVBKF-NG and DVBKF-SPSA are close to those in the centralized architecture. It is also clear that the PCRLB in distributed architecture is slightly bigger than that in centralized architecture, because only a part of the sensors are used in distributed architecture. However the computational cost in distributed architecture is much smaller than that in the centralized one. Table 4 and Table 5 give the computational cost comparison of the algorithms in centralized architecture and distributed architecture mentioned above.

The 3σ bound is another evaluation of target tracking accuracy. A solid line indicates the estimation error of associated algorithm. A dashed line indicates the 3σ error of the algorithm which presented as a solid line with the same color. Figure 7 presents the comparison of the 3σ bound. It is seen that the algorithms with NG or SPSA have smaller 3σ bound in range than CEKF and DEKF. The radical velocities 3σ of the proposed algorithms with NG and SPSA are slightly bigger than that of CEKF at some scans in Figure 7b. But we can observe that the radical velocities 3σ of the proposed algorithms are robust for the maneuvering target tracking both in centralized fusion architecture and distributed fusion architecture. The comparisons of 3σ error of estimated position and velocity in coordinates are given in Figure 8 and Figure 9, respectively. A solid line indicates the estimation error. A dashed line indicates the 3σ error of the algorithm which presented as a solid line with the same color. The quantitative 3σ comparison is given in Table 6 and Table 7.

As mentioned above, the minimization of KLD between variational distribution and posterior distribution is equivalent to maximization variational ELBO. To observe the changes in ELBO and KLD with iteration clearly, we present Figure 10 and Figure 11 to illustrate the normalized ELBO and KLD of the proposed DVBKF-NG and DVBKF-SPSA, respectively. In Figure 10, each line denotes the ELBO in one scan with 200 iterations, and each line in Figure 11 denotes KLD in one scan with 200 iterations. The results in Figure 10 and Figure 11 verify our standpoints by the following fact: the ELBO curves increase with the number of iterations, corresponding to a decrease in the KLD.

## 7. Discussion

In this paper, we address the problem of improving the accuracy for maneuvering target tracking in sensor networks. Two kinds of optimization methods, NG and SPSA, are introduced to maximize the distributed ELBO where the joint likelihood is partitioned approximately into several simple marginal likelihoods by variational mean field, formulating the algorithms of DVBKF-NG and DVBKF-SPSA. Moreover, the performance metrics, both of ELBO and PCRLB, are presented over different iterative indexes. In addition, a maneuvering target tracking scenario over a sensor network is given to verify the performance of the proposed algorithms. From the view of fusion architecture, centralized fusion architecture has lower RMSE curves and PCRLB curves than the distributed one. From the view of optimization methods, the simulation results show that the RMSE curves of the proposed algorithms are better than those which do not use any optimization methods.

For future work, we plan to adopt VB for the robust estimation of sensor networks. For example, outliers always lead to heavy-tailed and asymmetric distributions which are apt to lead to large estimation errors. It is expected for novel methods to mitigate the adverse influence and VB in which an unsolvable distribution can be approximated by a parameterized distribution is an appropriate method at this point. We also plan to develop VB for multiple passive sensor placement. For example, in a bearings-only target tracking system, tracking accuracy is highly dependent on the locations of the bearings-only sensors. Therefore, it is desirable to schedule the sensors’ moving trajectories in a way to achieve a minimized tracking error at a future time.

## Figures and Tables

**Figure 1 entropy-25-01235-f001:**
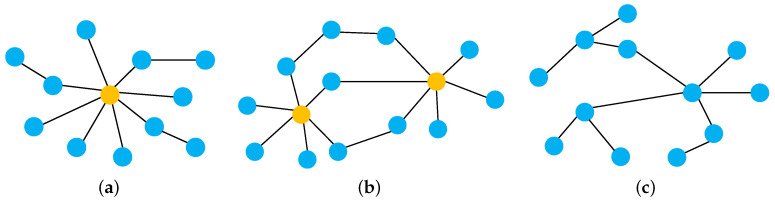
The architectures of sensor networks. (**a**) Centralized sensor network architectures. (**b**) Decentralized sensor network architectures. (**c**) Distributed sensor network architectures.

**Figure 2 entropy-25-01235-f002:**
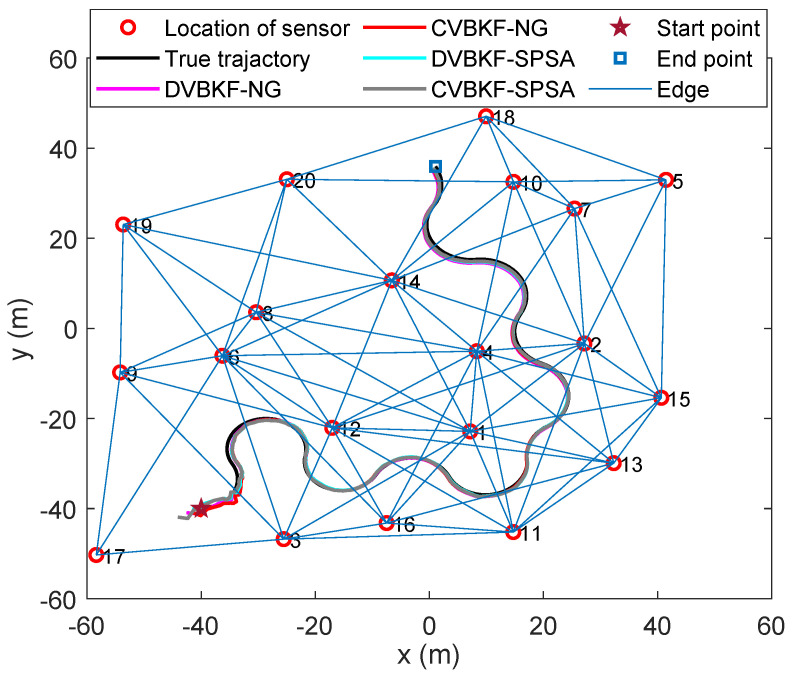
Sensor network scenario.

**Figure 3 entropy-25-01235-f003:**
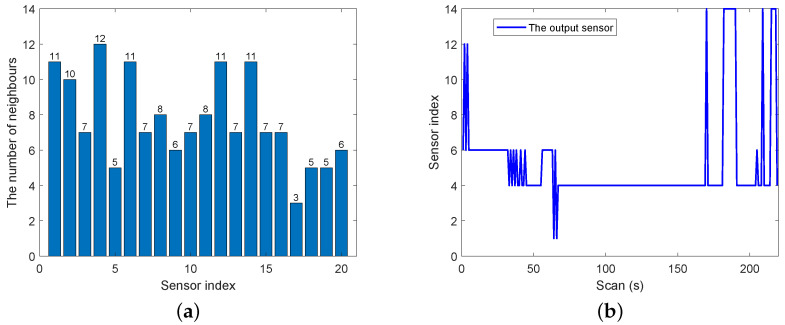
(**a**) The number of the neighbors of each sensor. (**b**) The output sensor against scan index.

**Figure 4 entropy-25-01235-f004:**
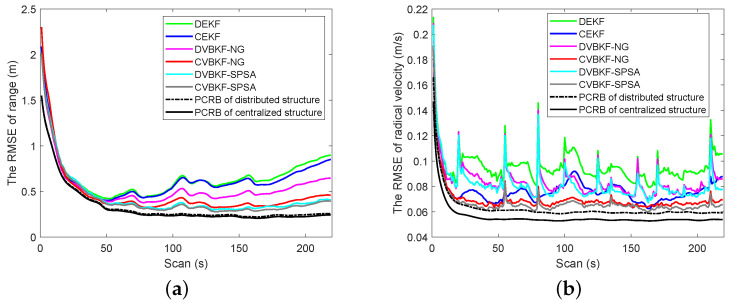
The comparison of PCRLB and RMSE. (**a**) The RMSE of range. (**b**) The RMSE of velocity.

**Figure 5 entropy-25-01235-f005:**
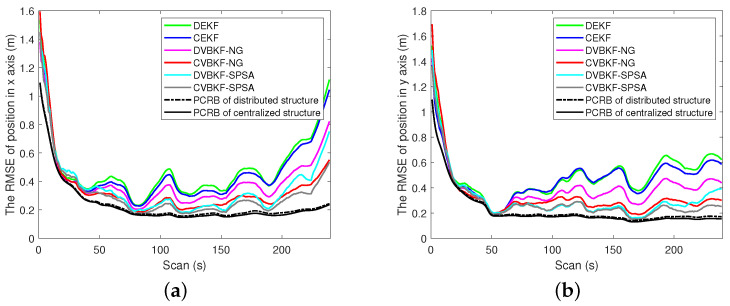
The comparison of the PCRLB and RMSE of estimated position in coordinates. (**a**) The RMSE on x-axis. (**b**) The RMSE on y-axis.

**Figure 6 entropy-25-01235-f006:**
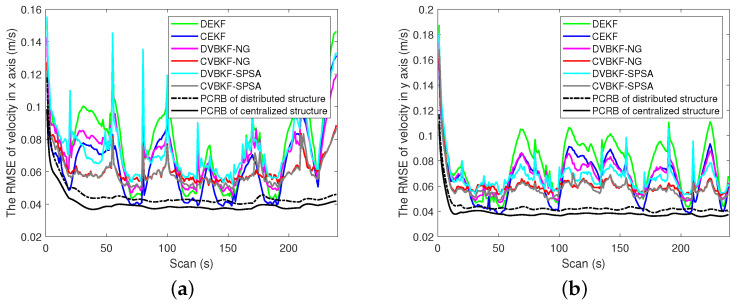
The comparison of the PCRLB and RMSE of estimated velocity in coordinates. (**a**) The RMSE on x-axis. (**b**) The RMSE on y-axis.

**Figure 7 entropy-25-01235-f007:**
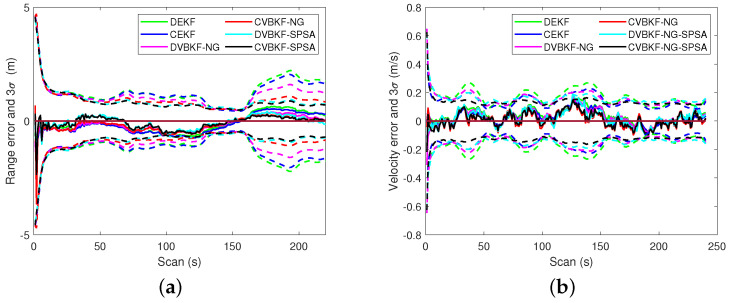
The 3σ comparison. (**a**) The 3σ of range. (**b**) The 3σ of velocity.

**Figure 8 entropy-25-01235-f008:**
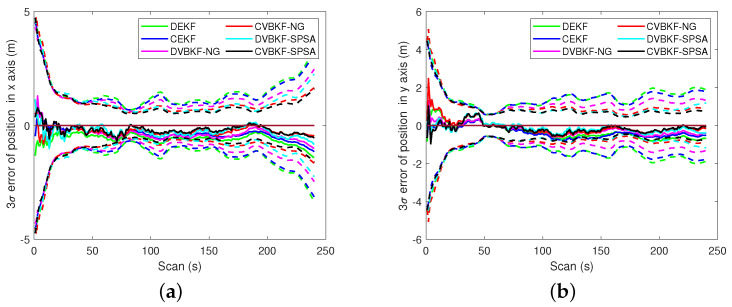
The 3σ comparison of estimated position. (**a**) The 3σ on x-axis. (**b**) The 3σ on y-axis.

**Figure 9 entropy-25-01235-f009:**
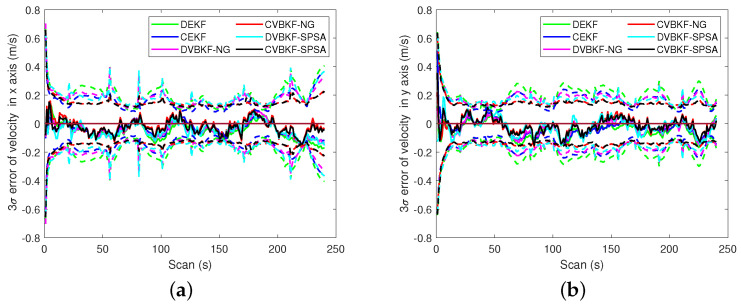
The 3σ comparison of estimated velocity. (**a**) The 3σ on x-axis. (**b**) The 3σ on y-axis.

**Figure 10 entropy-25-01235-f010:**
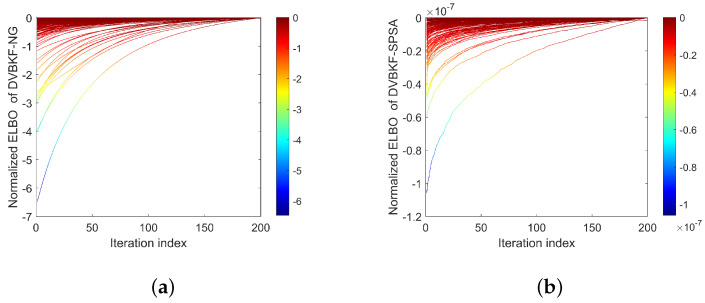
ELBO. (**a**) DVBKF-NG. (**b**) DVBKF-SPSA.

**Figure 11 entropy-25-01235-f011:**
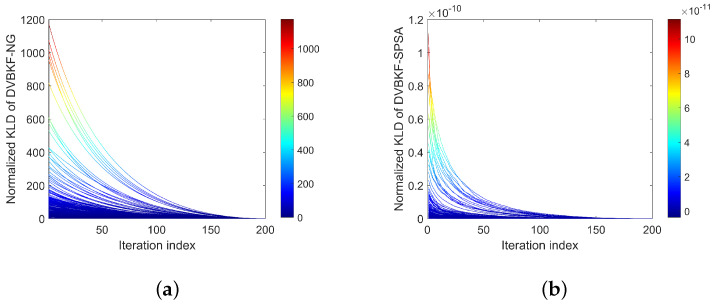
KLD. (**a**) DVBKF-NG. (**b**) DVBKF-SPSA.

**Table 1 entropy-25-01235-t001:** The values of parameters.

Parameter	a	b	c	α	γ	$c_{x_{k|k}^{i}}$	$c_{P_{k|k}^{i}}$	$a_{x_{k|k}^{i}}$	$a_{P_{k|k}^{i}}$
Value	0.01	20	100	1	0.16666701	$\frac{c}{i^{\gamma }}$	$0.01c_{x_{k|k}^{i}}$	$\frac{a}{{\left(i+b\right)}^{\alpha }}$	$0.001a_{x_{k|k}^{i}}$

**Table 2 entropy-25-01235-t002:** The comparison of mean RMSE in centralized architecture.

Algorithm	CEKF	CVBKF-NG	CVBKF-SPSA	PCRLB
Position (m)	0.6285	0.4636	0.4212	0.3411
Velocity (m/s)	0.0762	0.0695	0.0667	0.0627

**Table 3 entropy-25-01235-t003:** The comparison of mean RMSE in distributed architecture.

Algorithm	DEKF	DVBKF-NG	DVBKF-SPSA	PCRLB
Position (m)	0.6524	0.5373	0.4388	0.3277
Velocity (m/s)	0.0957	0.0837	0.0816	0.0560

**Table 4 entropy-25-01235-t004:** The comparison of computational cost in centralized architecture.

Algorithm	CEKF	CVBKF-NG	CVBKF-SPSA
Time (s)	0.0725	0.1827	0.3231

**Table 5 entropy-25-01235-t005:** The comparison of computational cost in distributed architecture.

Algorithm	DEKF	DVBKF-NG	DVBKF-SPSA
Time (s)	0.0614	0.1047	0.2102

**Table 6 entropy-25-01235-t006:** The comparison of 3σ centralized architecture.

Algorithm	CEKF	CVBKF-NG	CVBKF-SPSA
Position (m)	1.3833	1.0086	0.9196
Velocity (m/s)	0.1537	0.1550	0.1474

**Table 7 entropy-25-01235-t007:** The comparison of 3σ distributed architecture.

Algorithm	DEKF	DVBKF-NG	DVBKF-SPSA
Position (m)	1.4430	1.2050	0.9606
Velocity (m/s)	0.1835	0.1684	0.1714

## Data Availability

No new data were created or analyzed in this study. Data sharing is not applicable to this article.

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
