# Peer review of "Variational Bayesian Algorithms for Maneuvering Target Tracking with Nonlinear Measurements in Sensor Networks"

_entropy, 2023, doi:10.3390/e25081235_

Round 1

Reviewer 1 Report

The authors present an interesting topic of target tracking using the variable Bayes algorithm modeled with a particular case of the Kalman filter. The article is well structured and the algorithm is very elegantly described.

I recommend the authors that after chapter 7 Discussions they introduce the Conclusions chapter and reformulate chapter 7. Also in chapter 7 the authors can introduce a short comparison between the method chosen by them and similar methods reported in the specialized literature on the same topic.

Author Response

Dear reviewer,

Thank you for your comments.  We revised the part of Discussion, and added the comparison  between the proposed methods and the algorithms mentioned in simulations. The added part is given as follow:

From the view of fusion architecture, centralized fusion architecture has lower  RMSE curves and PCRLB curves than the distributed one. From the view of optimization methods,  the simulation results show that the RMSE curves of  proposed algorithms are better than those which doesn’t use any optimization methods.

Reviewer 2 Report

The paper discusses a method to estimate a moving target position and speed by a network of radar-like sensors. A variational Bayesian Kalman filtering algorithm is used to process a series of measurements made by these sensors. Unfortunately there is a poor correlation between the method and its application, so the text of the paper looks like a mixture of ideas. The main problems are:

1. Theoretical sections 2-4 are very general and they give no idea to the reader that the results of the measurement are the range and the speed. ‘x’ is defined just as a measurement. E.g. it is not clear if the velocity can be used to update the coordinate within this model.

2. In section 6 the model of the sensor is not described at all. Judging by the context I can guess that each sensor is something like omnidirectional Doppler-aware radar. I.e. each sensor send pings and it can estimate (1) a range to the target by the ping travel time, (2) target’s radial velocity according to Doppler effect, but (3) not the bearing. And each sensor receives echoes only for its own pings. However I can imagine different models after reading a caption ‘Doppler-only measurement’. E.g. target is equipped with a beacon, and the sensors are passive thus they measure only radial velocity, but not the range.

3. Many physical mistakes between the lines 324-327. Please check and rewrite the passage more clearly.

4. It is not clear why does the authors use the RMS error of the radial velocity to evaluate the network performance. The radial velocity is an output of a single sensor. The network is capable to estimate such universal values as x, y, Vx and Vy.

Comment on style: I suggest combining fig. 1, fig. 2 and fig. 3 into a single figure with fig 1a, 1b and 1c inside it.

Author Response

Dear reviewer,

Thank you very much for your careful work,we have given the responses as follow:

  1. Answer for comment 1:

In section 2, we formulate the problem of state estimation for maneuvering target tracking in sensor network with Doppler radars. Aiming to solve it, a distributed variational Bayesian optimization framework is proposed in section 3. On this basis, we proposed two different specific methods, namely as DVBKF-NG and DVBKF-SPSA, to implement target state in section 4.

In this manuscript, ‘x’ is target state and represented as $x_{k}=[x_{k}^p~x_{k}^v]^{\rm T}$, where $x_{k}^p=[x~y]^{\rm T}$ and $x_{k}^v=[\dot{x}~\dot{y}]^{\rm T}$ denote the true position and velocity of target. ‘z’ is measurement and represented in simulation section, both of them are defined in macro sense, are not as one state or one measurement. In the simulation, the state of velocity is updated in coordinate.

  1. Answer for comment 2:

In section 6, equations (49) and (50) represent the measurement model of the sensors, where (49) is the mapping from state to measurement and represents the Doppler shift between the target and each of sensors. Since the radial velocity of target is related to Doppler shift, we update it by using the velocity in coordinate. Equation (50) represents the associated time-vary measurement variance because of target motion.

  1. Answer for comment 3:

Thank you very much for your careful reading, we have revised it. Furthermore, we have had the manuscript polished with a professional assistance in writing.

  1. Answer for comment 4:

Thank you very much for your comment.

Because radial velocity is necessary to calculate Doppler shift, a low RMSE curve of the estimation of radial velocity is expected.  As explained above, we update it by using the velocity in coordinate.

We also revised the fig. 1, fig. 2 and fig. 3, combining them into a single figure with fig 1a, 1b and 1c inside it.

Thank you again.

Kind Regards,

Yumei Hu

Round 2

Reviewer 2 Report

Dear authors, I am afraid you have not fully addressed my comments/

May be this is due to my comments were hard to understand. If that was something like that, then I am really sorry for that. However the problems are:

1) Clear, but, please, put this information somewhere on the top of your paper.

2) Please give physical meaning in a simple form prior to exploiting sophisticated math.

3) You assume SNR = Ps/Pn and Ps = r^-4 i.e. per unit energy, you don’t put the transmitted power in it and Pn = kT/2 which is of certain dimension. Combining these two things in one ratio seems like a mistake. Finally, SNR should be dimensionless or dB, but that is not true in your case.

I am not going to the details that using temperature-induced noise in the denominator of SNR is too optimistic.

4) We don’t understand to which sensor this radial velocity belongs too, since you consider a group of sensors.

Moreover to that, a possible user of your system will be happy to see a radial velocity as an output of a single sensor, by won’t be happy to see this as an output of the whole system. (Vx,Vy) would be more helpful, and their error estimates are of  great interest.

Thus, I suggest one more turn of the major revision for your paper.

Author Response

Dear reviewer,

Thank you for your comments.
1. Answer for comment 1:
We put the information on page 13 from line 321 to 324.

2. Answer for comment 2:
Combined with the third comment, the physical meaning of equations (49) and (50) are given on page 13, from line 330 to 343.  We also presented the relationship between measurement noise covariance and the range from radar to target in Appendix C.   Thank you very much for your useful comments.

3. Answer for comment 2:
please check the information from line 330 to 343.

4. Thank you for your comment. Acturally, we expect to obtain the speed of target, and confused speed and velocity in expression.  The RMSE of estimated position and velocity in coordinate are given in figure 5 and figure 6. The $3 \sigma$ error of estimated position and velocity in coordinate are given in figure 8 and figure 9. We also presented the text of the figures on page 15 and page 16.

Thank you for your work again.

Best Regards,

Yumei Hu

Round 3

Reviewer 2 Report

The readers are more likely to understant the paper correctly in this version. I think it worth to be published. A couple of notes:

a) line 451: "For Gaussian noise Pn=kT/2 .." -> in fact that is true for the thermal noise (Johnson–Nyquist noise). Gaussian noise is a widder contept, not nessesary related to the temperature.

b) Futher polishing of the text is advised bu non nessesary.